# Using the Cocrystal Approach as a Promising Drug Delivery System to Enhance the Dissolution and Bioavailability of Formononetin Using an Imidazole Coformer

**DOI:** 10.3390/ph17111444

**Published:** 2024-10-28

**Authors:** Jongyeob Kim, Sohyeon Lim, Minseon Kim, Eunmi Ban, Yongae Kim, Aeri Kim

**Affiliations:** 1College of Pharmacy, CHA University, Seongnam 13844, Republic of Korea; jongyub@chamc.co.kr (J.K.); sh31227@naver.com (S.L.); eunmi.ban@gmail.com (E.B.); 2Department of Chemistry, Hankuk University of Foreign Studies, Yongin 17035, Republic of Korea; alstjs032@naver.com

**Keywords:** formononetin, cocrystal, imidazole, solubility, physical stability

## Abstract

**Background:** Natural isoflavones are recognized for their diverse pharmacological activities; however, their low aqueous solubility presents a significant challenge for further development. Here, we aimed to develop a cocrystal of formononetin (FMN) to improve its solubility. **Methods:** The formononetin-imidazole (FMN-IMD) cocrystal was prepared using liquid-assisted grinding method. The prepared cocrystal was identified through a thermal analysis of physical mixtures with various coformers. FTIR and solid-state NMR confirmed the presence of hydrogen bonds and π-π interactions in the FMN-IMD cocrystal. **Results:** The solubility of FMN-IMD was two to three times higher than that of crystalline FMN. The FMN-IMD cocrystal showed a 4.93-fold increase in the Cmax value and a 3.58-fold increase in the AUC compared to FMN after oral administration in rats. There were no changes in the PXRD of the FMN-IMD cocrystal after six months of storage at 40 °C. **Conclusions:** Thus, the FMN-IMD cocrystal is proposed as an effective solid form for oral delivery, offering enhanced solubility and physical stability.

## 1. Introduction

Formononetin (FMN, Figure 1) is an isoflavone with various pharmacological activities, including anti-cancer [1] and neuroprotective [2,3] activities and the ability to counteract intracellular ROS overproduction [4]. Almatroodi SA et al. presented various mechanisms for the anti-cancer action of formononetin in their recent review paper [5]. These mechanisms include the modulation of cell signaling pathways, such as cell cycle arrest, the activation of tumor suppressor genes, the inhibition of angiogenesis, the induction of apoptosis, and the inhibition of inflammation and invasion. The regulation of different signal transduction pathways involves the signal transducer and activator of transcription 3 (STAT 3), phosphatidyl inositol 3 kinase/protein kinase B (PI3K/Akt), and mitogen activating protein kinase (MAPK) signaling pathways. Singh L. et al. summarized the key molecular pathways modulated by formononetin that confer neuroprotection, such as downregulating the level of ROS and proinflammatory cytokines [3]. Chen H et al. provided both in vitro and in vivo evidence of the therapeutic potential of FMN as a promising candidate for the treatment of type 1 diabetes mellitus [4]. Despite previous reports on the pharmacological activities of FMN and its potential clinical applications, the poor solubility of FMN has been a major challenge in further development [1,2,3,6]. For example, Qnais E. et al. used a dimethylsulfoxide solution of formononetin to evaluate its effects on hyperglycaemia after oral administration in diabetic mice [6]. For the intraperitoneal injection of FMN to measure its efficacy in oxaliplatin-induced peripheral neuropathy, Fang Y. et al. used a solvent composed of 5% glucose solution and 2% dimethylsulfoxide in corn oil [7]. The poor aqueous solubility of a pharmacologically active compound is a significant challenge in research and development, particularly affecting the bioavailability of oral administration products. Previous reports on drug delivery systems, such as nanoparticles, nanofibers, or phytosomes, have demonstrated a significant improvement in the solubility of formononetin [8,9,10]. However, these systems are not suitable for processing into conventional solid dosage forms, such as capsules or tablets. Cocrystallization has emerged as a promising approach to enhance the pharmaceutical properties of poorly soluble compounds, including solubility, stability, and processibility [11,12,13,14,15,16]. According to FDA guidelines, a cocrystal consists of an active pharmaceutical ingredient (API) and coformer that are bound together in a specific stoichiometry ratio within the same crystal lattice [17]. Cocrystals are distinguished from salts because, unlike salts, the components that co-exist in the cocrystal lattice with a defined stoichiometry interact nonionically. These components interact through non-covalent interactions, including hydrogen bonding and π-π interactions [18,19].

In this study, we aimed to identify a cocrystal of FMN and conducted coformer screening for FMN using a thermal analysis of physical mixtures of FMN with various coformers. We examined the physicochemical properties of the resulting cocrystal and elucidated the role of intermolecular interactions between FMN and the coformer. Additionally, we assessed the oral pharmacokinetics in rats to compare the bioavailability of FMN and its cocrystal.

## 2. Results and Discussion

### 2.1. Coformer Screening for Cocrystal Formation Through Thermal Analysis

Coformer screening was carried out through a thermal analysis of physical mixtures of FMN and various coformers. An exothermic peak followed by a new endothermic peak appears on DSC for two components capable of forming cocrystals [13,20,21]. Among various physical mixtures, only the mixture of FMN and IMD showed an exothermic peak attributable to cocrystal formation (Figure 1). DSC profiles of other mixtures are shown in the Appendix A. The first endothermic peak from IMD melting in the physical mixture of FMN and IMD was immediately followed by an exothermic peak, which was then followed by the second endothermic peak. A sharper exothermic peak was observed when the heating rate was lowered to 2 °C/min. The sharpness of the exothermic peak can depend on the heating rate [21]. The second endothermic peak in the physical mixture of FMN and IMD was close to the melting point of the FMN-IMD cocrystal, suggesting that it occurred due to the melting of the cocrystal formed during the heating of the FMN and IMD physical mixture. There were no thermal events recorded during cooling after the melting of the cocrystal, indicating a super-cooled melt without any recrystallization (Figure 1). The second heating runs of both the physical mixture and FMN-IMD cocrystal showed a broad exothermic peak followed by a sharp endothermic peak, which can be assigned to the recrystallization and melting of FMN itself.

FMN-IMD cocrystallization in the FMN and IMD physical mixture was corroborated by HSM. Video recordings of these experiments are provided in the Appendix A. Figure 2 shows the representative polarized light microscopic images during the heating cycle of HSM.

Both IMD and FMN remained crystalline at up to 70 °C, showing their birefringence. After the melting of IMD, liquid IMD interacted with FMN at 95 °C to form a cocrystal by melt crystallization [13]. The cocrystal formation was complete at 102 °C, and the cocrystal started melting at 130 °C. The recrystallization of FMN at 265 °C and melting at 285 °C observed during HSM are consistent with the broad exothermic peak followed by a sharp endothermic peak at the corresponding temperatures during the second heating run of the FMN and IMD physical mixture or cocrystal (Figure 1).

### 2.2. Characterization of FMN Cocrystal

#### 2.2.1. Thermal Analysis

The DSC and TGA curves of FMN, IMD, and the FMN-IMD cocrystal are shown in Figure 3a and Figure 3b, respectively.

The FMN-IMD cocrystal showed a single endothermic peak at 134 °C, distinctive from those of FMN at 260 °C and IMD at 92 °C. This endothermic peak is consistent with the cocrystal melting observed during HSM (Figure 2d). The TGA profile of IMD indicates that the broad endothermic peaks observed in DSC after melting endotherm is due to the vaporization of IMD. The stoichiometric ratio of the cocrystal can be confirmed through a TGA [11]. The observed weight loss near the melting temperature of the cocrystal was 19.6%, which closely matches the theoretical weight percentage of IMD (20.2%) in the 1:1 mole ratio of the FMN-IMD cocrystal. No thermal events were recorded during the cooling phase after the cocrystal melted, indicating that the melt remained super-cooled without undergoing recrystallization. During the second heating run of both the physical mixture and cocrystal, there appeared a broad exothermic peak followed by a sharp endothermic peak at 255.8 or 254.0 °C (Figure 1). These thermal events can be interpreted as FMN recrystallization after cocrystal melting, IMD evaporation, and subsequent melting of recrystallized FMN. Likewise, the recrystallization at 265 °C and melting at 285 °C observed during HSM can also be attributed to FMN recrystallization and melting. The cocrystal can enhance the dissolution rates of poorly soluble compounds because their crystalline lattice energy can be lower than the original compounds. The large difference in melting points between FMN (260 °C) and the FMN-IMD cocrystal (134 °C) provides evidence of such lowering of the crystalline energy by cocrystallization.

#### 2.2.2. PXRD Analysis

PXRD patterns of FMN, IMD, and the FMN-IMD cocrystal are shown in Figure 4. PXRD is an effective method of distinguishing solid forms with different internal structures by comparing peak positions [22]. Therefore, PXRD has been used as one of the diagnostic analyses for cocrystal formation [13,23,24].

Table 1 summarizes their characteristic peaks. The presence of new, distinct peaks in the FMN-IMD cocrystal, such as those at 8.8°, 11.4°, 13.7°, 18.3°, 19.3°, and 20.8°, which differ from those of FMN and IMD, provides evidence of the cocrystal’s formation.

#### 2.2.3. ATR-FTIR Analysis

FTIR was used to elucidate the intermolecular interactions between FMN and IMD. The FTIR spectra of FMN, IMD, and the FMN-IMD cocrystal are shown in Figure 5, and the major peaks are summarized in Table 2.

FMN has a stretching C=O band at 1638 cm^−1^. Upon cocrystal formation, the C=O band shifted to 1626 cm^−1^ due to hydrogen bond formation. In the cocrystal spectrum, broad 2400 cm^−1^ and 1865 cm^−1^ bands appeared. These broad bands are indicative of the intermolecular O-H∙∙∙N hydrogen bond between FMN and IMD [25,26]. An aromatic compound shows skeleton vibration peaks at 1606/1567/1512 cm^−1^ [27]. Upon FMN-IMD cocrystal formation, these peaks shifted to 1607/1581/1510 cm^−1^.

#### 2.2.4. ^13^C-Solid-State NMR Analysis

Attempts to obtain a single crystal of an FMN-IMD cocrystal failed. As an alternative analytical tool, solid-state NMR (SSNMR) is useful when a single crystal is unavailable, providing information about differences between the cocrystal and starting materials [28,29,30,31]. Figure 6 shows the ^13^C-solid-state NMR spectra of FMN, IMD, and the FMN-IMD cocrystal.

The chemical shift difference between the starting materials and the cocrystal can be used to corroborate the cocrystal’s formation. Chemical shifts of FMN, IMD, and the FMN-IMD cocrystal are summarized in Table 3. Cocrystal formation resulted in the chemical shift of FMN C4 at 175.8ppm to 177.5 ppm and a 164.9 ppm shift of C7 to 165.8 ppm. In addition, chemical shifts of ring B in FMN (C1′, C2′, C3′, C4′, C5′, and C6′) at 123.1, 124.1, 107.4, 157.7, 117.1, and 127 ppm shifted to 125, 125, 110.9, 161.7, 120.5, and 129.9 ppm. These distinctive changes may be due to π-π interactions between the FMN ring B and the IMD ring.

The intermolecular interactions in the FMN-IMD cocrystal based on SSNMR are consistent with those of the ATR-FTIR results. The intermolecular interactions deduced from these two spectral analyses are hydrogen bonds involving O-H∙∙∙N and C=O and π-π interactions between the FMN ring and IMD (Figure 7).

### 2.3. Powder Dissolution Experiment

Bio-relevant media, FaSSGF and FaSSIF, were used for powder dissolution tests to mimic in vivo environments. The FMN-IMD cocrystal showed a faster dissolution rate and higher concentration at the 6-h time point than the crystalline FMN in both FaSSGF and FaSSIF (Figure 8).

Cocrystals can enhance the dissolution rates of poorly soluble compounds because their crystalline lattice energy can be lower than the original compounds. The large difference in the melting points between FMN (260 °C) and the FMN-IMD cocrystal (134 °C) provides evidence of such lowering of the crystalline energy by cocrystallization. Supersaturating formulations such as cocrystals oftentimes show a solubility decrease after quickly reaching supersaturation due to phase transformation to a less soluble hydrate [13,32]. For example, Park et al. reported a “parachute effect” during the powder dissolution study of an emodin/nicotinamide cocrystal, which was attributed to a phase change due to a less soluble emodin hydrate in water [13]. Zhang et al. reported a rapid decrease in the solubility of flavonoid cocrystals after an initial peak concentration and the maintenance of elevated solubility compared to the flavonoid itself [33]. They attributed this initial solubility decrease and yet elevated equilibrium solubility compared to those of flavonoids to soluble complex formation between the flavonoids and coformers in the solution. Similar profiles can be found in [34]. Alvani A. and Shayanfar A. describe in their review paper the importance of the solution stability of cocrystals and the effects of various factors, such as polymers, surfactants, biorelevant media, organic solvents, and pH [35]. In the present study, there was no apparent decrease in the solubility in bio-relevant media, as shown in Figure 8, after the rapid dissolution of the cocrystal. The concentration remained steady during 6 h experiments, suggesting no phase transformation of FMN in the test media of FaSSGF and FaSSIF.

When the FMN-IMD cocrystal’s solubility was measured in the test media containing PVP K30, its solubility was enhanced further (Figure 8). This could be due to previously reported solubilizing effects of PVP K30 [36,37] or its specific interaction with FMN. The following rat pharmacokinetic study was carried out with samples suspended in PVP solution.

### 2.4. Rat Oral Pharmacokinetics

Oral PK profiles of FMN and the FMN-IMD cocrystal were compared to corroborate our hypothesis that the FMN-IMD cocrystal would improve the oral absorption of FMN. The study was carried out in rats at a 20 mg FMN/kg dose. The plasma concentration profiles and the PK parameters are presented in Figure 9 and Table 4, respectively.

The FMN-IMD cocrystal exhibited a 4.93-fold increase in Cmax and a 3.58-fold increase in the AUC compared to crystalline FMN. The results, consistent with the enhanced solubility of the FMN-IMD cocrystal, suggest that it is an effective solid form of FMN for further development.

### 2.5. Physical Stability of Cocrystal

PXRD data of the cocrystal samples after six months of storage at 4 and 40 °C are shown in Figure 10. They remained the same for six months with characteristic peaks at 8.8°, 11.4°, 13.7°, 18.3°, 19.3°, and 20.8°, as shown in Figure 4, confirming its physical stability [38,39,40].

## 3. Materials and Methods

### 3.1. Materials

FMN was purchased from Shaanxi Yi An Biological Technology Co., Ltd. (Xi’an, China). Imidazole, 4-hydroxybenzophenone, and formic acid (internal standard for pharmacokinetic analysis) were purchased from Sigma Aldrich (St. Louis, MO, USA). PVP K30 was purchased from BASF (Ludwigshafen, Germany). Nicotinamide, nicotinic acid, maleic acid, citric acid, aspartic acid, ascorbic acid, imidazole (IMD), L-proline, acetamide, malic acid, L-histidine, theobromine, 4-aminobenzoic acid, and theophylline were used for coformer screening. All other chemicals and solvents were of analytical or chromatographic grade.

### 3.2. Coformer Screening by Thermal Analysis

Physical mixtures of FMN and various coformers (Nicotinamide, nicotinic acid, maleic acid, citric acid, aspartic acid, ascorbic acid, and imidazole (IMD): Sigma-Aldrich, St. Louis, MO, USA; L-proline, acetamide, malic acid, L-histidine, theobromine, 4-aminobenzoic acid, and theophylline: TCI Chemicals, Tokyo, Japan) in a 1:1 molar ratio were prepared by mixing them for 1 min with a vortex mixer. Differential scanning calorimetry (DSC) was employed to analyze the physical mixtures at heating rates of 10 or 2 °C/min. DSC profiles were examined for the appearance of an exothermic peak or a new endothermic peak. Hot stage microscopy (HSM) was used to observe cocrystallization in a physical mixture of FMN and IMD at a heating rate of 3 °C/min. Physical mixture samples placed on a hot stage (Linkam Scientific Instruments Ltd., Redhill, UK) were observed under a polarized light microscope (BA310 POL, Motic Inc., Ltd., Hong Kong) for phase changes upon heating.

### 3.3. Preparation of FMN Cocrystal

FMN-IMD cocrystal was prepared by a liquid-assisted grinding method. Typically, 400 mg of FMN and 101.5 mg of IMD (1:1 molar ratio) were ground for 30 min using a pestle and mortar with 60 μL methanol. Samples were recovered in glass vials and stored at 4 °C until further analysis.

### 3.4. Characterization of Cocrystal

#### 3.4.1. Differential Scanning Calorimetry (DSC)

DSC measurements of samples were performed using the TA-Q20 instrument (TA Instruments, New Castle, DE, USA). Samples were placed in Tzero Pan and Lid and heated at 10 °C/min under nitrogen purging gas at 50 mL/min. Additional DSC run at 2 °C/min was carried out to see the effect of the heating rate on the exothermic peak.

#### 3.4.2. Thermogravimetric Analysis (TGA)

TGA of samples was performed using the TA-Q50 instrument (TA Instruments, USA). Samples were loaded into a platinum pan and heated at a rate of 10 °C/min under nitrogen purging gas at 50 mL/min.

#### 3.4.3. Polarized Light Microscopy and Hot Stage Microscope (HSM)

Powder samples were observed under a polarized light microscope (BA310 POL, Motic Inc., Ltd., Hong Kong) for birefringence. Hot stage microscopy (HSM) was used to observe phase changes under Motic polarized light microscope upon heating on a hot stage (Linkam Scientific Instruments Ltd., Redhill, UK).

#### 3.4.4. Powder X-Ray Diffraction (PXRD)

Powder samples were loaded on the PXRD sample holders and characterized by PXRD (SmartLab, Rigaku, Tokyo, Japan) at 40 kV and 40 mA (Cu target) with a scan step size of 0.02° over the 3–40° 2θ range with a scan speed of 3 °/min.

#### 3.4.5. Attenuated Total Reflectance Fourier Transform Infrared Spectroscopy (ATR-FTIR)

FTIR spectra were obtained using FTIR (Shimadzu, IRSpirit, Kyoto, Japan) with the MIRacle single reflection horizontal ATR accessory (PIKE Technologies, Fitchburg, WI, USA). A few mg of samples were loaded onto the accessory plate for analysis. A total of 100 scans were collected in wavelengths 4000–650 cm^−1^.

#### 3.4.6. ^13^C-Solid-State NMR

The ^13^C solid-state cross-polarization magic angle spinning (CPMAS) NMR spectra were obtained using Bruker Avance III HD 400 MHz spectrometer, with 9.4 T AscendTM standard-bore magnet (Bruker Biospin, Ettlingen, Germany) equipped with 2.5 mm Bruker MAS probe. The spinning rate for FMN, IMD, and the FMN-IMD cocrystal were 18 kHz, 10 kHz, and 20 kHz, respectively, and the number of scans were 2048, 5120, and 2048, respectively. All spectra were obtained at room temperature with a 3 ms contact time and a 5 s delay time and processed in Bruker Topspin 4.1.1 (Bruker Biospin, Germany).

### 3.5. Powder Dissolution Experiment

Powder samples were sieved through a 300 µm sieve. FaSSGF and FaSSIF were prepared according to the literature, except pepsin was not included in FaSSGF [41]. Samples equivalent to 5 mg of FMN were added to 20 mL glass vials containing 10 mL of FaSSGF or FaSSIF. Sample vials were placed in a shaking incubator (SIF-6000R, Jeio Tech, Daejeon, Republic of Korea) at 37 °C. Aliquots (1 mL) were collected at designated time points and filtered through a 0.45 µm PVDF filter and diluted with the HPLC mobile phase for HPLC analysis. Additional test for cocrystal was carried out in the presence of PVP K30 (2.5 mg/mL) in FaSSGF or FaSSIF.

#### HPLC Method for Powder Dissolution Experiment

FMN quantitative analysis was carried out using a Shimadzu system (Japan) equipped with a C8 column (5 μm, 4.6 mm I.D., and 150 mm length, Waters, Milford, MA, USA) at room temperature. The mobile phase was composed of methanol/water (60:40, *v*/*v*). Samples were injected in a volume of 10 μL with a total flow rate set at 1.0 mL/min. Detection was carried out at a UV wavelength of 254 nm. The area of the standard was used for calibration by linear regression analysis. The calibration curve (R^2^ > 0.999) showed good linearity in the range of 0.1~50 µg/mL.

### 3.6. Rat Oral Pharmacokinetics

#### 3.6.1. Pharmacokinetic Studies

The experiments were carried out in accordance with the Guidelines for Animal Experimentation of Nonclinical Center, Chaon Co. Ltd. (Yongin, Republic of Korea) and approved by the Animal Ethics Committee of the institution (CE23665) on 13 February 2024. Male Sprague−Dawley rats (230−300 g, 8 weeks) were obtained from Dae Han Bio Link Co., Ltd. (Eumseong-gun, Republic of Korea). All rats had free access to food and water in a controlled environment for one week. The rats were fasted overnight and randomized into two groups to receive FMN or FMN-IMD cocrystal suspension in 1% PVP-K30 solution. After stirring for 5 min at 37 °C, an appropriate volume (about 200 μL) of suspension was delivered via oral gavage at a dosage of 20 mg/kg body weight. Approximately 300 μL blood samples were obtained from tail veins and placed into tubes pretreated with sodium citrate before and after 0.25, 0.5, 1, 2, 4, 8, and 24 h of dosing. Blood samples were immediately centrifuged at 5000 rpm for 10 min. All the plasma samples were stored at −80 °C until analysis. FMN from plasma samples were prepared for analysis by liquid–liquid extraction using methyl tert-butyl ether according to the following steps: 60 µL internal standard (IS) solution (10 ng/mL in 50% methanol) was added to 70 µL plasma sample and 3 mL acetonitrile was placed in a 15 mL tube. The sample solution was vortexed for 10 min followed by centrifugation for 5 min at 4000 rpm. The upper organic layer was transferred to a clean test tube and evaporated on a heating block at 50 °C under nitrogen. The residue was dissolved in the 50% acetonitrile (900 µL, in 0.1% formic acid) and mixed on a vortex mixer for 30 s. Resulting sample solution was transferred to auto-sampler vials, and 5 µL aliquot was injected into the LC/MS/MS system for analysis of FMN by a validated method as described below.

The pharmacokinetic data were analyzed by Winnlene (China Pharmaceutical University, Nanjing, China), using the noncompartmental method to calculate the peak plasma concentration (Cmax), the time to reach Cmax (Tmax), and the area under the plasma concentration−time curve (AUC).

#### 3.6.2. LC/MS/MS Determination of FMN in Plasma

FMN quantitative analysis was carried out using Agilent 1200 system (Agilent, Waldbronn, Germany) with triple quadrupole mass spectrometry API5000 (ABSciex, Vaughan, ON, Canada) operated in negative electrospray ionization using MRM mode. A Luna C8(2) column (2.1 mm I.D., 100 mm length, 5 μm; Phenomenex, Torrance, CA, USA) was maintained at 40 °C. Gradient elution was carried out at a flow rate of 0.3 mL/min using mobile phase A (distilled water) and mobile phase B (acetonitrile). The gradient applied was as follows: from 0 to 2.5 min, 50% B; from 2.6 to 7.0 min, 95% B; and from 7.1 to 11.5 min, 50% B. Sample volumes of 5 μL were injected. The fragmentation transitions were *m*/*z* 267.1 → 252.0 for FMN and *m*/*z* 197.0 → 92.0 for FMN and IS, respectively. Optimized instrument settings specific for FMN and IS were as follows: curtain gas, 20 psi; ion source gas 1, 50 psi; ion source gas 2, 50 psi; ion spray voltage, −4500 V; turbo heater temperature, 650 °C. The precursor ions of FMN and IS were formed using declustering potentials of −90 and −40 V, respectively, and their precursor ions were fragmented at collision energies of −28 and −42 eV by collision-activated dissociation with nitrogen at a pressure setting of 5 (arbitrary units). Both quadrupoles were maintained at unit resolution. The ratio of the standard to IS peak area was used for calibration by linear regression analysis. The calibration curve (R^2^ > 0.999) showed good linearity in the range of 0.1~20 ng/mL.

### 3.7. Physical Stability Test

Powder samples in closed glass vials were stored at 4 and 40 °C. They were then analyzed by PXRD for crystallization at given time points. PXRD measurement conditions were the same as described above.

## 4. Conclusions

Selecting an optimal solid form is crucial to advancing pharmacologically promising compounds for further development. In this study, we aimed to develop a cocrystal of FMN with enhanced dissolution. Thermal screening successfully identified imidazole (IMD) as a coformer. The FMN-IMD cocrystal, with enhanced dissolution, improved oral absorption in rats and was stable for six months at an elevated storage temperature. Thus, the FMN-IMD cocrystal is proposed as a promising solid form of FMN for further development.

## Data Availability

The original contributions presented in this study are included in the article/Appendix A; further inquiries can be directed to the corresponding author.

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
