# Peer review of "Using the Cocrystal Approach as a Promising Drug Delivery System to Enhance the Dissolution and Bioavailability of Formononetin Using an Imidazole Coformer"

_pharmaceuticals, 2024, doi:10.3390/ph17111444_

Round 1
Reviewer 1 Report
Comments and Suggestions for Authors
In the current article (Cocrystal as a supersaturating delivery system for formononetin), authors prepared cocrystals to enhance dissolution and oral bioavailability of formononetin. The perform enough tests for the prepared formulations. However, I am facing a main issue while reading the manuscript. The writing is very weak, and the introduction and discussion do not present enough information. Therefore, the manuscript cannot be considered for publication at this stage.
Please find below the main issue that could help you improve it.
1. Title: Cocrystal as a supersaturating delivery system for formononetin
Why do you add supersaturating to your title?
Please make the title more attractive and present the work that was performed.
Here are some suggested titles:
a) Cocrystal Approach as a Promising Drug Delivery System to Enhance Dissolution and Bioavailability of Formononetin Using Imidazole Coformer
b) Cocrystal Approach: Enhance Dissolution and Bioavailability of Formononetin Using Imidazole Coformer
2. Introduction: Please revise all the parts of the introduction and improve them.
The introduction part gives a clear indication of the topic. In pharmaceutical formulations, three main areas (drug, drug delivery system, and disease) need to be covered. In addition, write a paragraph about the previously published technology to improve the dissolution of formononetin if it is present.
3. Line 41 (2.1. Coformer screening for cocrystal formation through thermal analysis):
Please provide the DSC curve of FMN and IMD in Figure 1.
4. Line 105: (2.2.2. PXRD analysis)
Please make a clear description of the obtained results and make a good discussion of the present results.
Please check the accuracy of the present results. The data of FMN and FMN-IMD cocrystal) presented in Table 1 was reversed by mistake.
5. Lines 183 to 186: Why was PVP K30 added to dissolution media? Is there any reference for this condition? surfactant line tween 80 and sodium lauryl sulfate are usually added to mimic the inv-vivo sink condition. If the cocrystal is suspended in PVP K30, you should perform an experiment for pure API suspended in PVP K30.
Also, in Figure 8, the PVP abbreviation was used with cocrystal FMN-IMD-PVK30. This was added to the media, not the formulation. So, please correct this or explain.
6. Line 205: In Figure 10, the PXRD of the cocrystal is completely different from that presented in Figure 4.
Please check and present all spectrums with similar metrics. For example, if you make similar smoothing for all spectrum at same condition.
7. Line 216 (3.2.Coformer screening by thermal analysis):
Please mention the used coformers in line 217: various coformers (xxxx, xxxx, and xxxx) in a 1: 1 molar ratio.
8. Line 224: (3.3. Preparation of FMN cocrystal)
were ground for 30min using a pestle and mortar with 60 μL methanol. Then, ………….
Please make a complete description of the preparation method till the storage of cocrystals for further analysis.
9. Line 229: (3.4.1. Differential Scanning Calorimetry (DSC))
2 or 10 ° C/min: please be more specific.
10. Line 260: (3.5. Solubility test)
I think this could be described as a dissolution test. Solubility is usually measured at a specific point in time, while dissolution is to study the rate of drug dissolution over a period of time.
11. Line 268: please make a new section for the UPLC method for drug analysis.
Please provide a clear description of the method developed. For example, mention the range of the calibration curve and column temperature.
12. Line 353: (references)
Please make enough citations from the literature in the introduction and discussion sections.
Please try to cite recent studies (in the last three years) if applicable.
Dear authors, please check all the comments above and take the time to improve the manuscript, present your data, and discuss the results you obtained very well.
Author Response
Please see see the attachment.

Reviewer 2 Report
Comments and Suggestions for Authors
In the submitted manuscript the Authors have obtained the co-crystal of formononetin and imidazole. Then, the obtained co-crystal was analyzed using physicochemical methods and in vivo. I appreciate the application of 13C CP MAS NMR, however it should be improved, as described below. The study is concise, which is good, however the presentation style should be improved too, especially the figures. I recommend major revisions, with the detailed comments listed below.
Major issues:
The Authors have forgotten to upload the supplementary materials. I’ll check them during the second round of review.
The Authors have performed the rat oral pharmacokinetics studies, however they haven’t stated whether the studied has been accepted by the local Ethics Committee, when exactly and what is the reference number of this agreement.
In the introduction, the Figure should be presented, showing the structures of both FMN and IMD.
Line 30, so, according to this definition, salts are also cocrystals?
Line 32, “with various pharmacological activities” such as?
Line 32, “poor solubility”, be more exact.
Lines 211-214, please list the manufacturers of those chemicals.
Line 257, why the number of scans differed?
Line 259, what was the repetition time?
Figures 1, 2, 3, 4 should be increased (size)
Figure 1, in the DSC of cocrystal there are two exothermic peaks. Pleas explain clearly both of them. One of them represents melting, and the second one?
Figure 2, please provide the scale.
According to line 62, the Figure 1 presents the results for “FMN-IMD cocrystal”. Are you sure? This DSC is different from the one presented in Figure 3.
Table 1, please provide the simulated (from cif) PXRD of FMN and IMD. You can find cif files in CCDC.
Line 129, this is not surprising as the authors have chosen the wrong method. Instead, they should have crystallize the co-crystal from solution.
Figure 6 is of very poor quality, it must be improved
Figure 6 a, either increase the number of scans, or, more likely, the recycle delay. You can barely see the peaks.
Figure 6b, how the peak assignment has been performed? By guessing?
Table 3, units (ppm) are missing
Figure 7, this is highly hypothetical and should be confirmed using molecular modeling (DFT) methods
Minor comments:
Why „Natural” with capital N ?
Line 366, “6.” should be removed.
Line 395, “19.” should be removed.
Round 2
Reviewer 1 Report
Comments and Suggestions for Authors
The authors did not make the necessary changes to improve the revised version of the manuscript (Cocrystal Approach as a Promising Drug Delivery System to Enhance Dissolution and Bioavailability of Formononetin Using Imidazole Coformer). Therefore, my recommendation is to reconsider after major revision. The main issue is the lack of scientific discussion of results and comparing these results with the literature. In addition, the introduction did not include sufficient background information for the presented results.
1. Please improve the introduction. This part needs to be scientific and give readers a clear idea about the manuscript.
a) Formononetin and its clinical application or potential
b) Previous studies to improve solubility, dissolution, and bioavailability of drugs. Below are some of the reported studies.
· Physicochemical properties and in vitro release of formononetin nano-particles by ultrasonic probe-assisted precipitation in four polar organic solvents
· Formononetin/methyl-β-cyclodextrin inclusion complex incorporated into electrospun polyvinyl-alcohol nanofibers: Enhanced water solubility and oral fast-dissolving property
· Formulation and Characterization of Phytosomes as Drug Delivery System of Formononetin: An Effective Anti-Osteoporotic Agent
c) The advantages of cocrystals over other drug delivery systems.
2. XRD results must be discussed more, and citations for similar findings, even with other drugs and conformers, must be provided.
3. Line 182: Authors mention dissolution rate. Why do the authors in line 293 mention the solubility test? This is a dissolution, not a solubility test.
Reviewer 2 Report
Comments and Suggestions for Authors
The Authors have revised their work, making current version acceptable.
Author Response
Thank you for your valuable comments and feedback during the review.
Round 3
Reviewer 1 Report
Comments and Suggestions for Authors
Thank you for addressing all comments. The manuscript can be published now.